# Tumor-Derived Extracellular Vesicles in Cancer Immunoediting and Their Potential as Oncoimmunotherapeutics

**DOI:** 10.3390/cancers15010082

**Published:** 2022-12-23

**Authors:** Meysam Najaflou, Mehdi Shahgolzari, Ahmad Yari Khosroushahi, Steven Fiering

**Affiliations:** 1Department of Medical Nanotechnology, Faculty of Advanced Medical Science, Tabriz University of Medical Sciences, Tabriz 51666-14766, Iran; 2Drug Applied Research Center, Tabriz University of Medical Sciences, Tabriz 51666-14766, Iran; 3Dental Research Center, Hamadan University of Medical Sciences, Hamadan 65175-4171, Iran; 4Department of Microbiology and Immunology, Geisel School of Medicine at Dartmouth, Hanover, NH 03755, USA; 5Dartmouth Cancer Center, Geisel School of Medicine at Dartmouth and Dartmouth-Hitchcock Medical Center, Lebanon, NH 03756, USA

**Keywords:** exosomes, immunoediting, cancer immunity, immune escape, immunosurveillance, tumor microenvironment

## Abstract

**Simple Summary:**

Tumor cell-derived extracellular vesicles (TEVs) are an important means of tumor communication with, and manipulation of, the patient’s physiology. TEVs influence the local tumor environment as well as the systemic conditions of the patient. Progressive changes in tumor interactions with the host immune system are defined as “immunoediting”. Here, we summarize TEV effects on the immune system during the stages of cancer immunoediting and outline the molecular and cellular characteristics of interactions that result in complete tumor regression versus tumor immune escape and progression. Generally, the cargo profile of TEVs naturally changes during immunoediting toward immunosuppression while different cell stress or treatment conditions can inhibit this process or even reverse it to immunostimulation by altering the TEVs cargos. Therefore, understanding potential immunotherapeutic properties and how they can be manipulated to treat cancer should be considered a new research approach in oncoimmunotherapy.

**Abstract:**

The tumor microenvironment (TME) within and around a tumor is a complex interacting mixture of tumor cells with various stromal cells, including endothelial cells, fibroblasts, and immune cells. In the early steps of tumor formation, the local microenvironment tends to oppose carcinogenesis, while with cancer progression, the microenvironment skews into a protumoral TME and the tumor influences stromal cells to provide tumor-supporting functions. The creation and development of cancer are dependent on escape from immune recognition predominantly by influencing stromal cells, particularly immune cells, to suppress antitumor immunity. This overall process is generally called immunoediting and has been categorized into three phases; elimination, equilibrium, and escape. Interaction of tumor cells with stromal cells in the TME is mediated generally by cell-to-cell contact, cytokines, growth factors, and extracellular vesicles (EVs). The least well studied are EVs (especially exosomes), which are nanoparticle-sized bilayer membrane vesicles released by many cell types that participate in cell/cell communication. EVs carry various proteins, nucleic acids, lipids, and small molecules that influence cells that ingest the EVs. Tumor-derived extracellular vesicles (TEVs) play a significant role in every stage of immunoediting, and their cargoes change from immune-activating in the early stages of immunoediting into immunosuppressing in the escape phase. In addition, their cargos change with different treatments or stress conditions and can be influenced to be more immune stimulatory against cancer. This review focuses on the emerging understanding of how TEVs affect the differentiation and effector functions of stromal cells and their role in immunoediting, from the early stages of immunoediting to immune escape. Consideration of how TEVs can be therapeutically utilized includes different treatments that can modify TEV to support cancer immunotherapy.

## 1. Introduction

Roughly 19.3 million new cancer cases are expected to be diagnosed in a year across the world. Before the recent establishment of immune-based cancer therapies, the immune system was generally considered to play a minor role in cancer biology. However, immunotherapies, specifically checkpoint blockade antibodies (CPB), are now established therapy for many cancers, with extensive ongoing research into new higher-impact immunotherapies [1,2]. It is now accepted that the immune system does recognize and attempts to eliminate cancer and an important question in oncology is: how do cancer cells evade the immune system? There is considerable information, but the process is complex and variable between tumor types and between patients, so the understanding is incomplete. The immune system acts as a double-edged that can control and eliminate tumors through or can help tumors progress through immunosuppression and support of angiogenesis [3,4].

The TME contains tumor cells, stromal cells including stromal fibroblasts, endothelial cells, and immune cells that can make up more than half of the overall tumor mass. These cells influence tumor cells and each other through complex interactions such as extracellular matrix (ECM) secretion, soluble factors, cell–cell communication, cytokines, chemokines, and inflammatory mediators [5,6,7]. Extracellular vesicles (EVs) are an important means of intercellular communication and include vesicles up to 1 μm with plasma membrane origin, and smaller lipid bilayer vesicles (30–100 nm), which according to the International Society for Extracellular Vesicles are called small extracellular vesicles (exosomes), that are cup-shaped or doughnut-shaped [8,9]. EVs carry and deliver membrane and cytosolic components, including proteins, lipids, and nucleic acids [10,11]. The physiological and pathological function of EVs depends on their contents and ability to deliver their cargoes. Like other secreted biologically active components, vesicular-based cell-to-cell communication does not require cell contact and can act over long distances [8]. Their internalization in target cells can be through direct fusion with the plasma membrane, endocytosis, phagocytosis, or through ligands on their surface binding to receptors on other cells [8]. Their compositions are associated with endosome biogenesis and parental cell type since EVs with different origins contain unique subsets of components with different cell type-associated functions [10].

The process of tumors interacting with the immune system has been characterized by three phases, elimination, equilibrium, and escape [12,13,14]. During the elimination phase, small tumors not clinically recognized are eliminated without awareness by anyone. During the equilibrium phase, tumors are held in check by immune pressure but not eliminated, and tumors in this stage are also generally not recognized clinically. Lastly, in the escape phase, tumors grow large and are clinically recognized as cancer [12]. It should be noted that most information about cancer comes from tumors in the escape phase, since before that they are small, not identified in humans, and generally hard to study in animal models.

During the elimination phase, the innate and adaptive immune systems work together to identify and eliminate transformed cells that have evaded genetic mechanisms that suppress malignancies. Tumor cells that survive the elimination phase replicate and enable the tumor to reach the equilibrium phase. Both innate and adaptive immunity appears to play a major role in limiting tumor progression in this phase and tumors are controlled, but not eliminated by the immune system. During equilibrium, the tumor is sufficiently immunosuppressive to avoid elimination but is unable to significantly expand. Ultimately in the escape phase, tumors with more robust immunosuppressive mechanisms overcome the immune system via different mechanisms, grow until they are clinically detectable, locally invade and generate metastases [12,13]. In each of the immunoediting phases, cancer-secreted factors interact with the immune system. Some of them can help the tumor grow by suppressing the functional immune cells, whereas others stimulate the immune cells to respond against cancer Figure 1 [12,13,15]. As noted below, EVs play a role in all three stages, and this role generally goes from an antitumor effect in the elimination phase to a protumor effect in the escape phase.

EVs from malignant cells are important mediators of malignant cell communication in the TME and beyond and may support cancer metastasis, angiogenesis, therapy resistance, and immunoregulation leading to resistance to immune surveillance [16,17,18]. Recent studies show that cancer cells use EVs to communicate with one another and with stromal and normal cells [8,10]. The cancer-mediated immunoregulation mechanisms and factors they influence include the expression of surface molecules such as PD-L1 by cancer cells and the recruitment of immunosuppressive cell types such as Tregs and myeloid-derived suppressor cells (MDSC). This review discusses the effects of TEV on the immune system in each phase of cancer immunoediting, their roles in immunoregulation in the TME, and their potential use in cancer immunotherapy.

## 2. Tumor-Derived Extracellular Vesicles and Tumor-Supportive Cells

During the initial phases of tumor formation, the local microenvironment has mostly anticancer effects [14], but as tumors progress, the healthy microenvironment changes into the TME, immune protection is lost, tumor growth continues, and the stromal cells are influenced by cancer cells to support tumorigenic functions [19,20,21,22]. Along with tumor growth, immunoediting proceeds step by step with the formation of the TME. Tumors that successfully escape the equilibrium phase alter the surrounding healthy microenvironment by manipulating surrounding cancer-associated fibroblasts (CAFs), Tumor-associated macrophages (TAMs), MDSCs, and other immune cells and by recruiting new TME cells that further suppress the immune response via cytokines/chemokines, cell-to-cell contact, growth factors, and EVs [4,23].

### 2.1. TEVs Modulate Macrophage Activity in TME

Macrophages are complex and plastic cells that adopt a range of phenotypes from strongly immune suppressive to strongly immune stimulatory depending on the environmental signals they receive. While there are no clear distinctions on this continuum, for convenience they are often divided into M1 subtypes with pro-inflammatory properties that express cytokines such as IFN-γ, TNF-α, CXCL-10, IL-12, and high amounts of nitric oxide (NOS), and M2 subtypes with an anti-inflammatory function that release anti-inflammatory cytokines such as IL-4, IL-10, IL-13 and express high amounts of arginase-1(ARG1), scavenger receptors, and mannose receptor [24]. TAMs, begin to shift phenotypic from M1 to M2 through macrophage polarization with exposure to tumor-derived factors and TEVs in the TME and hypoxic conditions and act as a bridge between the adaptive and innate immune systems [25,26]. The M2 cells manifest local supportive functions for the tumor [26,27]. Cancer-derived extracellular vesicles increase M2 polarization by activating signaling pathways such as STAT3, p38MAPK, NF-κB, ERK1/2, and PI3K/AKT [28,29,30] and reprogram M1 tending cells into the cancer-promoting M2 end of the macrophage phenotype spectrum. Table 1 outlines a variety of data on specific activities of TEV in mediating immunosuppression.

The phosphorylated STATs, besides supporting M2 polarization, augment the secretion of generally immune suppressive cytokines such as IL-6 and upregulate PD-L1 that directly suppresses effector T cells through PD-1 on the T cells [28,31,32,33]. Breast cancer cell-derived vesicular gp130 stimulates bone marrow-derived macrophages (BMDMs) to secrete IL-6 by transferring gp130 into BMDMs which results in phosphorylation of STAT3 causing macrophage polarization and IL-6 secretion [28]. Similar to gp130, vesicular Anx II coupling with STAT3 stimulates other signaling pathways in M2 polarization including the p38MAPK, and NF-κB pathways in macrophages, leading to augmented IL-6 and cancer progression [31]. Besides proteins, EVs deliver microRNAs such as miR-222, miR-29a-3p, and miR-146a-5p that also stimulate the STAT3 and the NF-B signaling pathways leading to M2 polarization [34]. Other microRNAs, miR-106b and miR-934, when transferred to macrophages via TEV, activate the PI3K/AKT signaling pathway, which also stimulates macrophages toward M2 polarization [30,35].

**Table 1 cancers-15-00082-t001:** Effect of tumor-derived extracellular vesicles on the macrophages in the tumor microenvironment.

Cancer Type	Cellular Source	Vesicular Cargo	The Main Result	Refs.
**Breast cancer**	MCF10A MCF10AT MCF10CA1a MDA-MB-231	Anx II	Activated NF-B, p38MAPK, and STAT3 pathways in macrophages, leading to increased IL-6 and TNF-α secretion	[31]
	C57BL/6 EO771	gp130	Caused macrophages to shift from a normal to a polarized phenotype such as TAM via activation of the IL-6 response pathway and STAT3.	[28]
	4T1	miR-125b-1-3p, miR-100-5p, and miR-183-5p	Inhibited the expression of PPP2CA, which could promote the release of pro-inflammatory cytokines such as IL-1b, IL-6, and TNF-a from macrophages stimulating tumor invasion.	[20]
	MDA-MB-231	Vesicular CD63 protein	Polarized and activated macrophages, in which CD206 (a marker for M2) was expressed more than NOS2 (a marker for M1).	[36]
**Prostate cancer**	PC3	miRNA Let-7b	Prostate-derived extracellular vesicles had more miRNA Let-7b than cellular miRNA Let-7b can lead to macrophage polarization.	[37]
**Lung cancer**	A549	Vesicular cargoes	Altered transcriptomic and bioenergetic profiles of macrophages, forced them to polarize to an M2 phenotype.	[38]
	NCI-H1437 NCI-H1792 NCI-H2087	miR-103a	Polarized monocytes toward immunosuppressive M2-type macrophages.	[39]
	A549 H1299	Vesicular cargoes	Enhanced the levels of MMP2, MMP9 CD163, TNF-, IL-8, IL-6, and IL-10 and decreased expression of iNOS which led macrophages to exhibit a dual M1/M2 phenotype	[40]
	A549 H1299	Vesicular PRPS2	Induced M2 polarization and led to drug resistance of cancer cells.	[41]
**Hepatocellular carcinoma (HCC)**	PLC/PRF/5	Long non-coding RNAs (lncRNA) TUC339	Caused macrophage polarization to be more immunosuppressive.	[42]
	Hepa1-6 H22	miR-146a-5p	Enhanced M2 polarization by triggering NF-B signaling and producing pro-inflammatory proteins	[34]
**Colorectal cancer(CRC)**	DLD-1	miR-145	Induced M2 polarization via upregulation of IL-10 and downregulation of HDAC11.	[43]
	Blood samples from CRC patients HCT116 HT29	miR-106b	Contributed to M2 polarization of macrophages via significant increase in the miR-106b level in macrophages. It directly suppressed programmed cell death 4 (PDCD4) at a post-transcription level that led to an activated PI3Kγ, AKT, and mTOR signaling cascade.	[35]
	Blood samples from CRC patients HCT-8 LoVo HT-29 Caco-2	miR-934	Induced M2 macrophage polarization by activating the PI3K/AKT signaling pathway and downregulating PTEN.	[30]
	CT-26 SW620	Cytoskeleton-centric proteins	In macrophages, caused cytoskeleton reorganization via promoting elongation and F-actin polarization.	[44]
	Blood samples from CRC patients HCT116 DLD-1 HT29	miR-1246	Reprogrammed macrophages into the cancer-promoting state after macrophage uptake.	[45]
	Blood samples from CRC patients DLD1 HCT116 Lovo SW480 SW620 HT29 CaR-1 RKO Colo205 Colo320DM	miR-203	Promoted M2 polarization, which modulated liver metastasis of colon cancer cells.	[46]
**Epithelial ovarian cancer**	SKOV3	miR-21-3p, miR-181d-5p, and miR-125b-5p	Promoted M2 macrophage polarization results in epithelial ovarian cancer cell proliferation and migration under hypoxic circumstances.	[47]
**Glioblastoma**	GSC20 GSC276 U87	Vesicular cargoes	The presence of phospho-STAT3 in TEVs switched monocytes toward the tumor-supportive M2 phenotype	[33]
	U87MG SBN19 U251	FasL, TRAIL, CTLA-4, CD39, and CD73	Promoted M2 polarization by activating the NF-κB pathway in macrophages	[48]
	U251	Vesicular cargo	Induced M2 polarization leading to tumor growth via promoting TAM Arginase-1+ exosome secretion	[49]
**Oral squamous cell carcinoma**	SCC-9 CAL-27	miR-29a-3p	Targeted macrophages directly, and activated p-STAT1 to promote M2 expression	[32]
	Cal-27	CMTM6	Delivered CMTM6 to macrophages and induced M2-like macrophage polarization by activating ERK1/2 signaling	[29]
**Ovarian cancer**	Blood samples from overian cancer patients Skov3 A2780	miR-222	Induced M2 polarization of macrophages by activating STAT3 pathway	[50]

### 2.2. TEVs Modulate Fibroblast Activity in TME

In the normal state, fibroblasts are activated during wound healing to help in the process by creating an extracellular matrix (ECM), which serves as a scaffold for other cells [51]. Cancer-associated fibroblasts (CAFs) resemble myofibroblasts and often make up the majority of the cancer stroma [51]. Unlike normal fibroblasts (NFs), CAFs create an excessive ECM and secrete pro-invasive molecules such as ECM-degrading proteases. Hence, CAFs support ECM remodeling, and invasion by producing various kinds of cytokines, growth factors, chemokines, and matrix-degradable enzymes [52,53,54]. The molecular mechanisms that convert normal fibroblasts (NFs) to CAFs in TME are not fully understood. MiRNAs have a major function in the transition and activation of fibroblasts, as evidenced by the fact that dysregulation and disruption of miR-1, 206, 31,214, 155, and 31 secretion leads to the differentiation of NFs to CAFs through modulating FOXO3a, vascular endothelial growth factor (VEGF), and CCL2 signaling [55,56]. Recent studies show that crucial miRNAs in TEVs promote the differentiation of NFs into CAFs [21,57,58,59] (Table 2). Ovarian cancer vesicular miR-630 transformed NFs into CAFs by activating the NF-κB and inhibiting the KLF6 pathway [60]. In another study, lung cancer vesicular miR-210 activated the JAK2/STAT3 pathway, and ten-eleven translocation 2 (TET2) promoted the transformation of NFs into CAF [61]. The JAK2/STAT3 pathway activated by miR-210 resulted in increased expression of some pro-angiogenic factors such as FGF2, MMP9, and VEGF. In addition, breast cancer vesicular proteins such as survivin and ITGB4 converted NFs into myofibroblasts by increasing superoxide dismutase 1 (SOD1) and lactate in CAFs in a BNIP3L-dependent manner [62]. In bladder cancer, the vesicular TGF-β protein activates the TGF-β pathway and triggers CAF differentiation by SMAD pathway activation [19,63].

### 2.3. TEVs Effect on MDSC Formation in TME

MDSCs normally protect the host from the damaging consequences of excessive immune activation in pathological conditions such as wound healing, but in the TME, MDSCs promote angiogenesis, invasion, and metastasis, as well as block antitumor immunity [68]. MDSCs can generate robust immunosuppressive responses via numerous pathways such as the release of reactive oxygen species (ROS), NO via iNOS, arginine depletion by arginase, secretion of immunosuppressive cytokines such as IL-10 and TGF-β, and stimulation of apoptosis of immune effector cells via the Fas ligand pathway [68,69,70,71]. Therefore, MDSCs are critical mediators in helping cancers evade the immune system. Immature myeloid cells (IMCs) can fail to differentiate under several pathologic conditions (infection, inflammation, and cancer) and develop the features of dysfunctional myeloid cells that include MDSCs through a variety of mechanisms involving numerous substances that accumulate in the TME. Several growth factors and interleukins such as GM-CSF and interleukin IL-6 promote the differentiation of IMCs into MDSCs via activating the STAT-3 signaling pathway [72,73]. MDSCs are a diverse category of IMCs with immunosuppressive characteristics and activities [69]. It also was reported that immature natural killer (NK) cells, can be converted to MDSC [74].

Various tumor-derived factors within or on TEVs surface induce MDSCs in vitro, including IL-1β, IL-6, IL-10, prostaglandin E2 (PGE2), TGF-β, stem cell factor (SCF), and VEGF [72,75] (Table 3). The most important vesicular mediators involved in the differentiation of IMCs into MDSCs include PD-L1, PGE2, TGF-β, and HSP70 [75,76,77,78]. STAT pathways participate since various TEV can differentiate bone marrow myeloid cells into MDSCs by activating STATs [79,80].

Among vesicular cargoes, vesicular PD-L1 enhanced MDSC and M2 formation in breast cancer and glioblastoma and stimulated MDSCs and nonclassical monocyte (NCM) differentiation [76,77]. In breast cancer, TEVs carrying PGE2 and TGF-β switched the differentiation of IMCs into MDSC and also stimulated MDSC expression of Cox2, IL-6, VEGF, and arginase-1 [75]. Vesicular miR-181a and miR-9 stimulated MDSC generation by inhibiting SOCS3 and PIAS3 (two major regulators in the JAK/STAT signaling pathway’s negative feedback loop) [80], and mIR-1246 in glioblastoma-derived EVs induced activation and differentiation of MDSCs via specificity phosphatase 3 (DUSP3) in an ERK-dependent manner [81].

**Table 3 cancers-15-00082-t003:** Effect of TEVs on immature myeloid cell differentiation to MDSC in the TME.

Cancer Type	Cellular Source	Vesicular Cargo	The Main Result	Refs.
**Breast cancer**	4T1 tumor model in BALB/c mice	PGE2 and TGF-β	Induced the differentiation of IMCs to MDSC expressing IL-6, Cox2, VEGF, and arginase-1.	[75]
	MCF-7 4T1 MDA-MB-231	PD-L1+	Boosted tumor growth and accumulation of MDSCs and M2 in the TME.	[76]
	4T1	Vesicular cargoes	Differentiated bone marrow cells into MDSCs	[79]
	4T1 tumor-bearing mice plasma 4T1	miR-181a and miR-9	Stimulated MDSC differentiation by inhibiting SOCS3 and PIAS3 (regulators of the JAK/STAT signaling pathway).	[80]
**Gastric cancer**	MKN-28 MKN-45 SGC-7901	Vesicular cargo	Increased frequency of MDSC, and decreased CD8+ T and NK cells.	[22]
**Renal cancer**	RenCa	HSP 70	Antigen-specific immunosuppression effect on CTL	[78]
**Glioblastoma**	Blood samples from glioma patients	miR-1246	Induced MDSCs via specificity phosphatase 3 (DUSP3) and ERK-dependent manner.	[81]
	P3 G422 GL261 U87	miR-29a	Increased MDSCs via interaction with high-mobility group box transcription factor 1 (Hbp1) and protein kinase cAMP-dependent type I regulatory subunit alpha (Prkar1a).	[82]
	Blood samples from glioma patient Human astrocytes supernatant	Vesicular cargo	Acting on MDSC, reduced T-cell immune response in an indirect manner.	[83]
	Blood samples from glioma patient	PD-L1	Induced immunosuppressive monocytes, including MDSCs and nonclassical monocytes.	[77]
**Lung cancer (LC)**	95D H292 H358	miR-21a	Induced MDSC expression by downregulation of the PDCD4 protein.	[84]

## 3. TEVs-Mediated Communication between Tumor and Immune Cells

TEV early in tumor development can stimulate antitumor immunity. The interaction between immune cells and cancer in TME is categorized into seven potential steps (Figure 2) [4] which is called the “cancer-immunity cycle”. In the right conditions, TEV from tumor cells can also support antitumor immunity. TEVs contain and transfer TAs and damage-associated molecular patterns (DAMPs) to innate immune cells, especially dendritic cells (Step 1) [85,86]. Tumor-derived EVs are a source of shared TAs for CTL cross-priming.

Dendritic cells (DCs) respond to TEVs carrying DAMPs and TAs, mature, and migrate to lymph nodes (Steps 1–2). The tumor antigen is cross-presented on MHC class I (MHC-I) in the lymph nodes where it activates naive CD8 T cells (Step 3). The activated effector T cells go to the tumor site (Step 4), penetrate the tumor tissue (Step 5), identify cancer cells by tumor antigens presented on MHC-I (Step 6), then attack and kill them (Step 7). One or more of these stages may be disrupted in many cancer patients, resulting in ineffective immune responses to cancer. Disruption at any stage of this cycle is caused by cancer cells and their secreted factors, including via TEVs [86,87]. This disruption and immune system suppression block antitumor immunity and support cancer progression.

### 3.1. Elimination Phase TEV Involvement

This phase has not been directly detected in vivo in humans since it occurs with very small tumors. The innate and adaptive immune systems collaborate to identify and eliminate tumors that have evaded intrinsic tumor suppressor mechanisms in developing tumors [14]. Cancer immunosurveillance is proposed to remove newly generated neoplastic cells that have the potential to develop tumors. The processes of how the immune system is alerted of the presence of primary tumor cells remain unknown. Among the possibilities, the generation of neoantigens by abnormal cells within the created inflammatory environment (via immune cell subsets, recognition molecules, and effector cytokines) results in the detection of nascent cancers, and the traditional warning signals such as IFNs are likely involved [88,89,90]. T cells are the primary immune cells that identify and eliminate tumor cells [88,91]. However, B cells and their antibodies also seem to play a role in recognizing and removing these cells [92]. IFN-γ has a direct anti-proliferative impact on tumors through the STAT1 pathway and causes the release of cytokines such as CXCL9, 10, and 11 that increase immune activation by recruiting effector T cells [93,94]. IFN alpha and beta (type I IFNs) also play an important role in activating CD103^+^ DCs to cross-present tumor antigens [15,95].

Physical characteristics of the tumor environment such as hypoxia can cause tumor cell death, potentially resulting in the release of DAMPs such as Heat shock proteins (HSPs) and high mobility group box 1 (HMGB1), which act as ligands for Toll-like receptors on innate immune cells [85]. EVs can carry TAs, interferon, and DAMPs that stimulate immunological responses against tumors [96,97]. EVs carried CEA and HER2 TAAs that triggered immune responses and improved anti-tumor responses in vivo [98]. The release of tumor antigens and EVs can be altered under various TME situations. For example, an acidic microenvironment, quite common for tumors, increased the number of secreted EVs [99].

TEVs can play a key role in NK cell activation, DC maturation, and CD8^+^ effector T-cell development [100,101]. TEVs may also carry surface proteins derived from cancer cells which promote the uptake of TEVs by DCs. There are reports supporting LFA-1/CD54 and mannose-rich C-type lectin receptor interactions as enabling TEV uptake by DCs [102,103]. Uptake of TEVs by DCs enhanced DC expression of co-stimulatory receptors such as CD80, CD86, and also MHC II expression and boosted interferon and cytokine production along with DC maturation [104,105,106]. Breast cancer cells generated EVs that convey dsDNA to DCs, causing IFN alpha and beta expression in a STING-dependent manner and elevation of costimulatory molecules in DCs [107].

Furthermore, TEVs carry molecules that promoted CD8^+^ T-cell activation and enhanced tumor cytotoxic T lymphocyte (CTL) responses in vivo in mice [105,106,108]. EVs generated from brain tumors were delivered to mice on days 7 and 14 post-tumor inoculation, stimulating antibody production and T-cell activation. Antitumor antibodies and T cells present at the time of tumor inoculation appear to have caused enough tumor cell death to generate further T-cell antitumor response [109].

#### Tumor-Derived Immunostimulatory Vesicular DAMPs

During an immunogenic cell death (ICD), cancer cells release danger signals (DAMPs) raising the immunogenicity of dying cancer cells [85,110,111,112]. ICD is more immune stimulatory than necrosis which can suppress immunological responses [113] and necrosis generally does not strongly stimulate CD8^+^ T-cell-dependent immune responses [114]. DAMPs are secreted as a result of endoplasmic reticulum (ER) stress induced by mitochondrial ROS, membrane-lipid peroxidation, and ER-directed ROS generation [115,116,117]. DAMPs can also be released during necroptosis, pyroptosis, and ferroptosis [118,119,120]. EVs from cancer cells can carry DAMPs including HSPs, HMGB1, histones, ATP, vesicular RNAs, and cell-free DNA inside or on the surface [121,122,123,124,125,126]. Interestingly, EVs with surface-bound HSP70 stimulate more helper T cells (Th1) and CTL than TEVs with cytoplasmic HSP70 inside EVs [126]. Hsp70-enriched TEVs elicited significant CD4^+^ Th1 immune responses and promoted the production of MHC class II molecules on antigen-presenting cells, leading to the elimination of cancer cells [127]. CD94^+^ NK cells in the presence of TEVs possessing membrane HSP70 released granzyme B [126,128] and expressed stimulating receptors such as the NKG2D, CD69, and NKp44 while also down-regulating inhibitory receptor CD94 [129].

### 3.2. Equilibrium Phase TEV Involvement

Molecular processes that initiate immune-mediated cancer dormancy/control, i.e., the equilibrium phase (EqP), are not well understood in part because this phase is hard to model and has been minimally characterized in humans [130]. Not surprisingly, when overall mechanisms are poorly understood, there is not much known about the involvement of EVs in the equilibrium phase. In the equilibrium phase, the adaptive effector functions and the resistance of the tumor are in a dynamic balance. There are clear indications that tumors in the escape phase having metastasized, can return to equilibrium following chemotherapy and be dormant for many years before relapse. This occurs in particular with metastatic breast tumors where metastatic cells stop proliferating but survive in a quiescent state [131]. The role, if any that the immune system plays in maintaining this dormancy is not clear.

In the EqP, TEVs may suppress different adaptive immune cell types through various mechanisms such as inhibiting effector cells such as CD8^+^ T cells and NK cells, suppressing DC maturation and activation, increasing M2 and TAM immune suppressive polarization, and stimulating CAF differentiation [64,132,133]. However, as we noted previously, TEV can also mediate tumor-suppressing signals. TEVs containing miR-23b derived from mesenchymal bone marrow cancer stem cells (CSC) can induce cancer dormancy via downregulation of the MARCKS gene that mediates breast cancer cells’ differentiation into CSCs through the Wnt-β-catenin pathway [134,135].

Considering PD-L1 and IFN-γ in the EqP of tumors is of interest for understanding the involvement of TEV and highlighting the complexity of molecular interactions. While IFN-γ supports CD8 T-cell effector function, IFN-γ stimulation also increases the quantity of PD-L1 on melanoma-released EVs that in turn suppressed the effector function of CD8^+^ T cells [136]. IFN-γ induced tumor dormancy when the interferon-gamma receptor 1 (IFNGR1) expression level was low but resulted in tumor elimination when it was high [137]. GW4869 treatment or Rab27a knockdown can inhibit vesicular-PD-L1 secretion, and significantly augment anti-PD-L1 therapeutic efficacy in 4T1 tumor growth [138]. Animal studies have shown that TEVs can also impair the production of interferons as well as decrease innate immune activity via EGFR- and MEKK2- dependent pathways [139].

### 3.3. Escape Phase TEV Involvement

Clinically recognized tumors have generally moved from equilibrium to escape. In the equilibrium phase, genome instability and accumulation of mutations in cancer cells over time leads to selection for low immunogenicity, expression of immune suppressive ligands, and escape from the immune system [140]. Tumors can eventually overcome antitumor immunity through mechanisms already mentioned, including tumor antigen editing, loss of MHC I expression, and expression of immune inhibitors such as PD-L1 [141,142,143] or suppressive mediators such as IL-10 [144], TGF-β [145], and TRAIL decoy receptors [146,147]. Recruitment and activation of immune-suppressing cells such as Tregs also contribute to escape [148].

#### 3.3.1. Effect of TEVs on Dendritic Cells

Maturation of DCs requires inflammation-related stimuli which stimulate the expression of co-stimulatory molecules such as CD86, CD80, and CD40. TEVs can modify or block the differentiation of immature myeloid cells (IMC) to DC or divert the DCs maturation from IMC to MDSC or M2 macrophage (Table 2 and Table 3) by interacting with bone marrow IMC and inducing the production of IL-6, and decreasing expression of CD83 and CD86, as reported for breast cancer, murine mammary adenocarcinoma, and melanoma [149,150]. TEVs also can disrupt DC maturation and T-cell immune response with HLA-G-associated mechanisms in renal cancer [133] (Table 4). Some vesicular proteins such as MALAT1 directly interact with DCs and induce DC autophagy, which decreases DC-mediated T-cell activation [151]. Furthermore, TEV-treated DCs were ineffective at inducing CD4^+^ T-cell proliferation and activation but promoted differentiation into Treg [152]. TEVs fatty acids can create immunologically dysfunctional DCs by increasing intracellular lipid content by activating the peroxisome proliferator-activated receptor (PPAR) resulting in extra fatty acid oxidation (FAO) which shifts the DCs’ metabolism toward oxidative phosphorylation of mitochondria and the disruption of the function of DCs [153,154,155]. It was reported that human prostate cancer-derived extracellular vesicles purified from cultured cells contained PGE2 and triggered the expression of CD73 and CD39 on DCs in vitro, resulting in the generation of adenosine from ATP and inhibition of TNF-α and IL-12-production which reduced T-cell activation [156].

HSP72 and HSP105 on the membrane of TEVs interact with TLR2 and TLR4 on DCs which induced IL-6 secretion by DCs that increased STAT3-dependent MMP-9 transcription activity in cancer cells resulting in tumor invasion [161]. Galectin-9 on glioblastoma-derived EVs binds to the TIM3 DCs receptor and inhibits antigen presentation by DCs, leading to disrupted antitumor immune responses of cytotoxic T cells [142]. Important DC receptors such as Tim-3 and galectin-9 [157] and SIRPα as the ligand for CD47 were up-regulated on the tumor cells’ membranes and derived TEV [143,164]. TLR4 on the DCs decreased after treatment with pancreatic cancer-derived vesicular miR-203 resulting in reduced expression of cytokines such as TNF-α and IL-12, subsequently reducing DC maturation and Th1 differentiation [125]. Besides the vesicular proteins, vesicular miRs also affect DC’s function. For example, miR-212-3p transferred to DCs by pancreatic cancer-derived extracellular vesicles suppressed regulatory factor X-associated protein (RFXAP), decreased MHC II expression, and reduced antigen presentation by DCs [165]. Table 4 summarizes reports of TEV impacts on DC.

#### 3.3.2. Effect of TEVs on T Cells

TEVs have a broad array of mechanisms by which they impact T cells. TEVs modify antitumor response by reducing T-cell viability, proliferation, and effector activities [166,167,168]. TEVs can disrupt T-cell effector function indirectly by blocking APC maturation [142,151,152] or directly by inhibiting activated CD8^+^ T-cell function, inducing CD8^+^ T-cell death through pro-apoptotic molecules (galectin-group proteins and FasL), promoting Treg expansion, and inducing T-cell exhaustion [169,170]. PD-L1 enriched glioblastoma-derived EVs perhaps surprisingly suppress monocytes rather than T-cells [77]. Nasopharyngeal carcinoma-derived vesicular galectin-9 induced apoptosis in CD4^+^ T cells via interaction with Tim-3 [171], as well as impairing T-cell function by interaction with TIM3 receptor on DCs in glioblastoma [142]. TEVs can carry pro-apoptotic Bax that induces apoptosis in CD8^+^T cells [172] and downregulates JAK3 expression which blocks CD8^+^ T-cell activation [167,173]. In Treg cell activation, both CD45 negative and positive EVs derived from plasma in head and neck cancer induced Treg differentiation of CD4 cells, but CD45(-) EVs also reduced CD8+ T-cell activation due to their higher adenosine concentrations [174]. EVs generated from multiple myeloma reduced the viability of CD4^+^ T cells and boosted the proliferation of Treg cells [175].

Vesicular PD-L1 promotes CD8^+^ T-cell apoptosis via PD-1/PD-L1 and PD-L1/CD80 signaling pathways [176], blocks T-cell activation in the draining lymph node in TRAMP- C2 prostate cancer mouse model [177,178], and reduces the proliferation of CD8^+^ T cells by decreasing IL- 2 and IFN-γ in the TME [136]. FasL on the TEVs decreased T-cell receptor (TCR) and CD3ζ expression in T cells leading to T-cell apoptosis [179], and melanoma-derived vesicular TNF downregulates TCR via redox signaling in T cells [180].

Pancreatic cancer cell EVs can stimulate p38 MAP kinase signaling in T lymphocytes that causes ER stress, which triggers the PERK–eIF2–ATF4–CHOP signaling cascade resulting in T-cell death [181]. Vesicular microRNAs in the serum of patients with nasopharyngeal carcinoma influenced T-cell differentiation and activation through suppression of the MAPK1 signaling pathway [182], while EVs with a high amount of miR-24–3 reduced CD4^+^ and CD8^+^ T-cell proliferation by targeting FGF11 [183]. In addition, mesothelioma cells’ EVs carrying TGF-β decreased proliferative response to IL-2 in T effector cells, but not in T-reg cells [184].

Vesicular galectin-1 plays a role in the induction of T-cell suppression [185]. TEVs also can induce T-cell exhaustion, by carrying inhibitory molecules, including PD-L1, CTLA- 4, TIM3, LAG3, and TIGIT [186,187]. miR-146a-5p and 14-3-3ζ in HCC-derived EVs induced T-cell exhaustion via activating M2-macrophages by inhibiting transcription factor SALL4 [30,188]. EVs carrying circRNA-002178 from patients’ serum with lung adenocarcinoma could boost PD-L1 production by sponging miR-34 in cancer cells, leading to CD8^+^T-cell exhaustion in vitro [132].

In addition, cancer patients’ plasma TEVs can prevent the activation of Th1 and Th17 lymphocytes and change them to immunosuppressive Treg phenotype cells [167,182]. The mutant KRAS gene is involved in the NSCLC-generated EVs-mediated transition of naive CD4^+^ T cells towards a FoxP3^+^ T-reg phenotype in a cytokine-independent manner in an NSCLC xenograft mouse model [189]. Table 5 summarizes reports on TEV suppressive effects on T cells.

#### 3.3.3. Effect of TEVs on NK Cells

NK cells play an important role in cancer immunosurveillance by expressing death-inducing ligands such as FasL, TRAIL and JAK/STAT pathway [197,198]. However, like most immune cells, the activation of NK cells is controlled by a complex balance of activating and inhibiting signals. Tumor cells trigger several activating receptors, such as NKG2D, natural cytotoxicity receptors (NCRs), and DNAX accessory molecule-1 (DNAM-1/CD226) [199]. Vesicular NKG2D, TGF-β, and MICA*008 suppress or downregulate the expression of NKG2D in both NK and CD8^+^ T cells resulting in decreasing cytotoxicity of these cells by reducing the expression of cytotoxic molecules [200,201,202,203,204,205,206].

## 4. Potentials of EVs in Cancer Therapy

TEVs have both immunostimulation and immunosuppression effects [96,97,161], but the potential has not yet been clinically utilized. Admittedly, TEV-focused therapy will be challenging. One challenge to using EVs in cancer immunotherapy is developing a system that provides uniform reagents for clinical use. However, a variety of preclinical studies illustrate the therapeutic potential of TEVs.

TEVs from tumors in the elimination phase stimulated immune cell responses against cancer development [101,104,128], while, not surprisingly, by the time tumor progression occurs, TEVs tend to suppress immune cells and support tumor immune escape. Three general cancer immunotherapy approaches involving TEV can be conceived: I) inhibition of TEV secretion II) increasing the immunostimulatory factors on TEVs’ surfaces, and III) using EVs as carriers in cancer vaccines.

To block the secretion of TEVs, the factors involved in their secretion, such as endosomal sorting complexes required for transport machinery (ESCRT), soluble NSF attachment protein receptor (SNARE), and Rab proteins (Rab11, Rab 27a, Rab 27b, and Rab 35) could be suppressed using drugs including Y27632, Imipramine, Calpeptin, Manumycin A, D-Pantethine, GW4869, and Simvastatin [207,208,209].

Cells under stress produce more immunostimulatory molecules on TEVs and secrete more EVs, which can be caused by treatment-induced stress [210]. Thus some cancer therapies can increase the production of immunostimulatory vesicular factors and this may help to disrupt and even overcome the process of usual tumor immunoediting [211,212]. Opposing increased immunostimulation from tumor EVs, treating with immune checkpoint inhibitors can boost the secretion of immunosuppressive EVs [136].

Hyperthermia is a useful cancer treatment and heat or other stress can modulate TEVs. Heat-stressed B lymphoma cells’ EVs possess more IL-6 and IL-17 stimulating molecules such as HSP90, HSP60, HSP70, CD40, and CD86, which can turn Tregs into Th 17 cells [211,213]. Heat stress boosts MHC-I expression on tumor cells [106] and generates TEVs equipped with chemokines such as CCL2,3,4,5, and CCL20 that functionally activate DC and T cells more strongly against tumors [214] thus stimulating “self-vaccination” [215]. Irradiated mouse breast cancer cells’ TEVs transmit dsDNA to DCs and induce DC to overexpress costimulatory molecules as well as STING-dependent type I IFN [107] and irradiated melanoma cells’ TEVs contained DC activation DAMPs such as HSP70, HMGB, and other stress-related proteins [124]. EVs from Melphalan (a genotoxic drug) treated myeloma cells can boost NK cell IFN-γ production via activating the NF-κB pathway in a TLR2/Hsp70-dependent manner [123]. IFN-γ treated cancer cells secreted a high amount of immune stimulatory TEVs that enhance the number of M1 macrophages by improving their capacity to ingest TEVs and promoting antibody production against cancer cells [216] as well as reducing Tregs and suppressing the expression of PD-L1, VEGF receptor 2, and IDO-1 [217]. In another strategy, modification of the vesicular contents by non-stress methods such as a lentiviral vector encoding two B7 costimulatory molecules (CD80, CD86) increased CD86 and CD80 expression in DCs and induced proliferation of CD4^+^ T cells, Th1 cytokine secretion, and CTL response [218].

TEVs have been studied as vaccine carriers and employed as immunogens for DC loading. Their immunogenicity in boosting DC-driven anti-tumor immunity was greater than tumor lysate, and they increased splenocyte proliferation and IL-2 release in mouse leukemia and melanoma cancer models [219,220]. In different syngeneic mouse models with large tumors, TEVs equipped with HMGB1 augment DC immunogenicity and elicit long-lasting antitumor immunity and tumor suppression [221]. Overall, EVs from a variety of cell types, including immune cells such as DCs and cancer cells, have the potential as a cancer vaccine and cancer immunotherapeutic [222] such as for colon cancer [2].

TEVs have potential use as drug carriers since they have an affinity for ingestion by cancer cells, are biocompatible and non-toxic with long half-lives in circulation, and their potential has been evaluated [223,224,225]. Some studies carried doxorubicin and in comparison to free doxorubicin, they boosted the therapeutic efficacy [224,225]. Additionally, TEVs carrying doxorubicin and paclitaxel crossed the blood–brain barrier (BBB) as part of in vivo studies [223]. This strategy of using TEVs to carry chemotherapy drugs as a cancer treatment has clinical potential and needs further study.

## 5. Conclusions

TEVs cargos are not static during cancer development; they change as tumors evolve and are stressed for various reasons. TEVs modulate immunostimulating or immunosuppressing effects against cancer cells by modifying immune cells during the tumor immunoediting phases [12]. TEVs play an immunostimulatory role in the early stages of immunoediting [8,12,96,101], are more immunosuppressive in the escape phase, and finally, at the late stages, they are more uniformly immune suppressive and play a major role in cancer immune escape [19,156,168]. Normally. the cargo profile of TEVs naturally changes in tumor immunoediting toward immunosuppression, while different cell stress or treatment conditions can inhibit this process or even reverse it to immunostimulation by altering the TEVs cargos profile. Therefore, to benefit from the therapeutic effects of EVs, the secretion of immunosuppression TEVs could be inhibited by disrupting the normal process of immunoediting. Generating EVs to apply therapeutically could be achieved by stimulating the release of immune-activating vesicular cargoes in vitro by using suitable treatment methods and subsequently using EVs in vivo as adjuvant therapy. Since TEV’s very diverse cargo profiles depend on the cancer cell condition, understanding the immunotherapeutic properties of TEVs to utilize against cancer should be considered a new research line in oncoimmunotherapy.

## Figures and Tables

**Figure 1 cancers-15-00082-f001:**
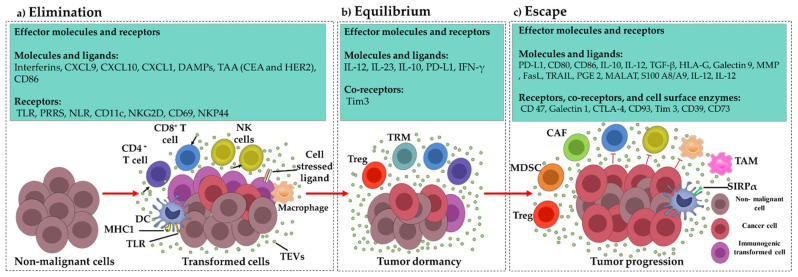
Cancer immunoediting and involved effector molecules and receptors: (**a**) During the elimination phase, the innate and adaptive immune systems work together to identify and eliminate transformed cells that have evaded genetic malignant cell suppression mechanisms. The generation of tumor antigens and extracellular vesicles by cancer cells trigger immunological responses against tumors that result in the detection and elimination of nascent cancer cells by the immune system (**b**) if they survive the elimination phase, tumors may reach the equilibrium phase. Adaptive and innate immunity limits tumor progression in this phase, therefore tumor growth is reduced, and surviving tumor cells are kept under control and possibly kept dormant by the immune system (**c**) Ultimately in the escape phase, tumors overcome the immune system by immune suppression through recruiting immunosuppressive cells, and suppressing effector immune cells via different mechanisms.

**Figure 2 cancers-15-00082-f002:**
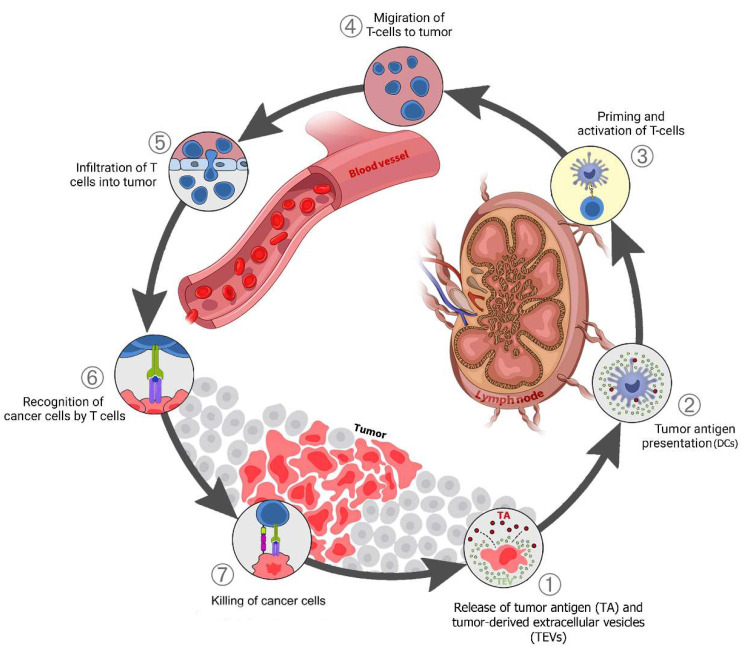
TEV in the cancer-immunity cycle: (**1**) Release of tumor antigens (TAs) along with tumor-derived extracellular vesicles (TEVs) that carry TA + DAMPs from dying cancer cells; (**2**) Presentation of TAs on the major histocompatibility complex (MHC) by dendritic cells; (**3**) T-cell receptor recognition of TAs on the MHC, leading to T-cell activation; (**4**) Migration of activated T cells to the tumors; (**5**) T-cell infiltration into the tumor; (**6**) Recognition of cancer antigens within the tumor; (**7**) Attack and killing of tumor cells.

**Table 2 cancers-15-00082-t002:** Effect of tumor-derived extracellular vesicles on fibroblast differentiation in the tumor microenvironment.

Cancer Type	Cellular Source	Vesicular Cargo	The Main Result	Refs.
**Triple-negative breast cancer**	MDA-MB-231 BT-20 MDA-MB-453 MCF7 BT-474 SK-BR-3	ITGB4 proteins	Enhanced mitophagy and lactate generation in CAFs in a BNIP3L-dependent manner.	[64]
	TNBC MDA-MB-231 MDA-MB-468	miR-9	Influenced the properties of NFs and promoted the switch to CAF, thereby leading to tumor growth.	[58]
**Ovarian cancer**	A2780 SKVO3	miR-630	Raised amounts of α-SMA and FAP in NFs resulted in the differentiation of NFs into CAFs via inhibiting KLF6 and activating the NFκB pathway.	[60]
**Bladder cancer**	RT4 T24 SW1710	TGF-β	Triggered the differentiation of fibroblasts to CAFs by SMAD pathway activation	[63]
**Colorectal cancer**	SW620 SW480	Vesicular cargo	Activated normal quiescent fibroblasts (α-SMA−, CAV+) into CAF-like fibroblasts (α-SMA+, CAV−) with pro-proliferative and pro-angiogenic features	[65]
**Lung cancer (LC)**	A549 H460	miR-210	Promoted the NFs transferring into CAFs via activating JAK2/STAT3 pathway, and ten-eleven translocation 2 (TET2)	[61]
**Melanoma**	B16BL6	eTGF-β	Triggered TGF-β signaling in HUVECs and differentiated them into CAFs	[19]
	B16F0	Gm26809	Stimulated conversion of fibroblast NIH3T3 cells into CAFs	[66]
	B16-F10	miR-21	Stimulated invasiveness of fibroblasts by increasing matrix metalloprotein (MMP) and down-regulation of tissue inhibitor of metalloproteinase 3 (TIMP3) expressions.	[59]
**Breast cancer**	MCF-7 MDA-MB-231	Vesicular survivin	Converted NFs into myofibroblasts by upregulating SOD1 and increased proliferation, EMT, and stemness.	[62]
**Head and neck squamous cell carcinoma (HNSCC)**	SAS HSC-3	Vesicular cargo	Convert normal fibroblasts into CAF-like cells and raised fibroblast proliferation, migration and activation of 11 signaling pathways (IL-6- and IL-17-related signaling)	[67]

**Table 4 cancers-15-00082-t004:** Effect of the tumor-derived extracellular vesicles on Dendritic cells.

Cancer Type.	Cellular Source	Vesicular Cargo	The Main Result	Refs.
**Prostate cancer**	DU145	PGE2	Triggered the expression of CD73 and then CD39 on DCs, resulting in inhibition of TNFα- and IL-12-production via an ATP-dependent manner	[156]
**NSCLC**	Blood samples from NSCLC patients	Galectin-9 and Tim-3	Interacted with TIM-3 on DCs	[157]
**Renal cancer**	CD105^+^ CSCs CD105^−^ TCs	HLA-G	Disrupted maturation of DCs and T-cell immune responses	[133]
**Glioblastoma**	CSF samples from glioma patients GL261 U87MG U118 MG	Galectin-9	Inhibited antigen recognition, processing, and presentation by interacting with TIM-3 on DCs	[142]
	Ascites of glioma patients	PD-L1	Impaired DCs maturation via formation of immunosuppressive monocytes	[77]
	Blood samples from glioma patients GSC20 GSC267 GSC17 MEC-1	Vesicular cargo	Skewed monocytes toward an immune suppressive phenotype and induced programmed PD-L1 expression on monocytes through STAT3 phosphorylation and TLR7-dependent manner	[33,158]
**Melanoma**	SKMEL28 A375 C32TG	S100, A8/A9	Inhibited DCs maturation and reduced expression of CD83, CD86, Th1 polarizing chemokines (such as Flt3L, IL-15), and migration chemokines (MIP-1α and MIP-1β)	[150]
	lymphatic fluid sample of melanoma patients ATCC	S100A9	Inhibited DCs maturation and prepared metastatic niche in lymph nodes	[159]
	B16-F0	TGF-β1	Increased mRNA levels of IL-4 and TGF-β1 which inhibited DCs’ maturation	[160]
	Blood samples from melanoma patients B16-F0	HSP72 and HSP105	Induced secretion of IL-6 from DCs via TLR4- and TLR2-dependent manner activating STAT3-dependent MMP 9 activity	[161]
**lymphocytic leukemia**	Blood samples from CLL patients	S100A8/A9	CD83, CD86, IL-12, and IL-15 expressions were all downregulated via activating the NFκB pathway	[162,163]
**lung carcinoma**	LLC	PD-L1	Myeloid precursor cells were unable to differentiate into CD11c+ DCs in the presence of vesicular PD-L1 and resulted in DCs death	[152]
	LLC A549	MALAT1	Inhibited DC function and T-cell proliferation and increased DC autophagy via AKT/mTOR Pathway	[151]
**Breast cancer**	MDA-MB-231 TS/A	Vesicular cargo	Inhibited the development of myeloid precursor cells into DCs by increasing IL-6 production and reducing CD83 and CD86 expression	[149]
	4T1	PD-L1	Myeloid precursor cells were unable to differentiate into CD11c+ DCs in the presence of vesicular PD-L1 and resulted in DC death	[152]
	Blood samples from melanoma patients 4T1	HSP72 and HSP105	Promoted DCs to IL-6 secretion in a TLR2- and TLR4-dependent manner which activated STAT3-dependent MMP 9 activity	[161]

**Table 5 cancers-15-00082-t005:** Effect of the tumor-derived extracellular vesicles on T cells.

Cancer Type	Cellular Source	Vesicular Cargo	Mechanism of Action	Refs.
**Ovarian cancer**	Ascites of ovarian patients OVCAR3 SKOV3 AD10	TGF-β1, IL-10	Increased IL-10, FasL, TGF-β1, CTLA-4, which promoted Treg proliferation, suppressor activity, and Treg cell survival.	[190]
	Blood samples from ovarian patients Ascites of ovarian patients OVCAR-3 AD10 A2780 Skov3 CaOv-3 MDAH2774 OvCa-14 OVP-10	Arginase-1	Inhibited antigen-specific T-cell proliferation	[191]
**Prostate cancer**	Pleural fluid samples of malignant pleural mesothelioma patients DU145 PC3	PGE2	T-cell inhibition was mediated through the adenosine A2A receptor	[192]
	DU145 PC3	TGF-β1	Skewed IL-2 responses in T cells and suppressed cytotoxicity	[184]
**Melanoma**	Blood samples from melanoma patients Blood samples from melanoma tumor-bearing mice WM1552C WM35 WM793 WM902B UACC-903 1205Lu WM9 WM164	PD-L1	Suppressed the function of CD8 T cells	[136]
**Colorectal cancer**	Blood samples from colorectal patients SW403 CRC28462 1869col	FasL, TRAIL	Induced T-cell apoptosis	[168]
	DLD-1 WiDr	TGF-β1	Induced differentiation of T cells to Treg-like cells via the TGF-β pathway while inactivating the SAPK signaling pathway	[193]
	Caco-2	Galectin- 1	Induced suppressor phenotype in human CD8+ T cells	[185]
**Head and neck** **cancer**	Tu167 SCC0209 HN60	Galectin- 1	Induced suppressor phenotype in human CD8+ T cells	[185]
	Blood samples from HNSCC patients	Vesicular cargo	Induced apoptosis in CD8+ T cells by converting CD4+ T cells to Treg	[174]
**Glioblastoma**	Blood samples from glioma patients UPN933 E3-2 E6-5	Vesicular cargo	Deactivated T cells by FasL-dependent mechanisms and inhibit secretion of IL-2	[194]
**Nasopharyngeal cancer (NPC)**	Blood samples from NPC patients Blood samples from NPC tumor-bearing mice C15 C17	Galectin- 9	Induced huge apoptosis in T cells via membrane receptor Tim-3	[171]
	Blood samples from NPC patients C15 C17	CCL20	Facilitated Treg recruitment and expansion that increased secretion of immunosuppressive cytokines (IL10, TGFB1)	[195]
	Blood samples from NPC patients TW03 C666 CNE2	miR- 24–3p	Blocked T-cell proliferation and Th1 and Th17 differentiation and promoted Treg induction via dephosphorylating ERK, STAT1, and STAT3 by reducing IL-2, IFNγ, and IL-17 secretion and phosphorylating STAT5 with increasing IL-6, IL-1β, and IL-10 secretion	[182,183]
**Oral squamous cell carcinoma(OSCC)**	SCC-9 SCC-4 CAL-27	HSP70	Altered development and cytotoxicity of T cells in an HSP70-dependent way via miR-21/PTEN/PD-L1 regulatory pathway	[170]
	Blood samples from OSCC patients PCI-13	FasL	Induced apoptotic pathways in T cells through triggering caspase-3 cleavage, the release of cytochrome c that led to disrupting mitochondrial membrane, and decreased TCR-ζ chain production	[172]
**Breast cancer**	MCF7	CD73, CD39	Inhibited T cells via the adenosine A2A receptor	[192]
	BT-474 MDA-MB-231	TGF-β1	Suppressed T-cell proliferation	[196]
**Lung cancer**	Blood samples from lung cancer patients A549 PC9 95D	circRNA- 002178	Enhanced PDL1 expression led to induced T-cell exhaustion	[132]
**Hepatocellular** **Carcinoma (HCC)**	Blood samples from HCC patients MHCC97H	14- 3- 3ζ	Inhibited the functions of T cells against cancer in the HCC microenvironment	[188]
	Hepa1-6 H22	SALL4/miR-146a- 5p	T cells were exhausted by reducing IFN-γ and TNF-α expression while increasing the expression of inhibitory receptors such as PD-1 and CTLA-4	[34]
**Pancreatic cancer**	BxPC-3 tdTomato-BxPC-3	Vesicular cargo	Induced ER stress-mediated apoptosis via activating the p38 MAP kinase signaling	[181]

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
