# Peer review of "Tumor-Derived Extracellular Vesicles in Cancer Immunoediting and Their Potential as Oncoimmunotherapeutics"

_cancers, 2022, doi:10.3390/cancers15010082_

Round 1

Reviewer 1 Report

The review “Tumor-Derived Exosomes in Cancer Immunoediting and Their Potential as Onco- Immunotherapeutics” summarizes reports on the immunostimulatory and immunosuppressive effects of tumor-derived extracellular vesicles (EVs). This topic is highly significant and could be of interest to a broad audience. However, in the current manuscript, several aspects are addressed in a superficial manner and lack precision, hence major revision is required. 

-          Authors use the terms “exosomes” and “tumor-derived exosomes” (TEXs) throughout the text. MISEV2018 guidelines (Thery C. et al, 2018) do not recommend using the term “exosome” unless the endosomal biogenesis pathway of the vesicles is clearly established. This certainly is not the case in the current paper, therefore I would strongly recommend using the term “extracellular vesicles” (EVs). 

-          Authors list many reports where specific EV cargo molecules have been shown to activate various signaling pathways and elicit changes in the physiology of recipient cells. However, I am not convinced that all the studies listed provide sufficiently strong evidence that the given cargo molecule indeed causes these effects. Hence, a deeper and more critical assessment of the evidence should be provided and some examples of how the causal relationships between the EV cargo and the effect in the recipient cells have been demonstrated should be included. 

-          The paragraph describing the principle of immunoediting is superficial and imprecise (lines 59-76). For example, the statement “Adaptive immunity limits tumor progression in this phase (equilibrium phase)” suggests that innate immunity does not have a role in the anti-tumor immune response in the equilibrium phase which is misleading. Furthermore, talking about immunoediting without citing the seminal papers of Robert D Schreiber, who proposed the concept and provided experimental evidence of immunoediting is not acceptable. 

-          Figure 1 also is not clear and precise. Classification of “cancer cells” vs “immunogenic transformed cells” is vague. What exactly is the difference between them? Effector molecules and receptors listed in the boxes should be classified according to their type or functions. Abbreviations should be spelled out in the legend. 

-          Tables 1, 2, 3, 4, and 5: cellular source of EVs should be included. 

-          Table 5: indicating “Plasma of patients” in the column “Cancer type” is not appropriate and should be replaced by the cancer types of patients from whom the plasma samples were collected.

Author Response

1. Authors use the terms “exosomes” and “tumor-derived exosomes” (TEXs) throughout the text. MISEV2018 guidelines (Thery C. et al, 2018) do not recommend using the term “exosome” unless the endosomal biogenesis pathway of the vesicles is clearly established. This certainly is not the case in the current paper, therefore I would strongly recommend using the term “extracellular vesicles” (EVs). 

  • Many thanks for the valuable comments. In the previous draft we tried to choose articles that have used the ‘Exosome’ term instead of small extracellular vesicles and have done the physical characteristics (below the 100 nm) and biochemical composition characteristics (CD63+/CD81+/CD9+). However, as you recommended and according to ISEV guidelines, we changed the exosome term to extracellular vesicles (EVs). We explained in the introduction that our point of EVs is small EVs in the exosome size range. The changing of nomenclature accounts for the vast majority of changes in the manuscript since the nomenclature pervades the manuscript.

2. Authors list many reports where specific EV cargo molecules have been shown to activate various signaling pathways and elicit changes in the physiology of recipient cells. However, I am not convinced that all the studies listed provide sufficiently strong evidence that the given cargo molecule indeed causes these effects. Hence, a deeper and more critical assessment of the evidence should be provided and some examples of how the causal relationships between the EV cargo and the effect in the recipient cells have been demonstrated should be included. 

  • We recognize that tumor immunology is complex and molecules that are generally pro- or anti-tumor could serve the opposite effector function in the right context. We have tried to summarize almost all the vesicular factors that interacted with the immune system by noting specific references. According to previous studies, many of these factors like PD-L1, when in EVs, indeed cause physiological changes in the recipient cells by known and demonstrated mechanisms. In this article, we tried to include the immunoediting function of these factors when they are in the EVs state as well. Differences of these factors in soluble and vesicular forms has been mentioned by suitable references (like ref 139) which indicated that vesicular cargoes, like PD-L1, have more immunoediting effect than soluble molecules.
  • We have expanded discussion of the factors with more evidence of their effects in the text. However, most of the involved factors with potential impact on immunoediting have been mentioned only in the tables briefly. We think it is useful to include them as potential examples of EV cargo that may impact immunoediting with the expectation on their intervention mechanisms and effects on the recipient cells in the future.

3. The paragraph describing the principle of immunoediting is superficial and imprecise (lines 59-76). For example, the statement “Adaptive immunity limits tumor progression in this phase (equilibrium phase)” suggests that innate immunity does not have a role in the anti-tumor immune response in the equilibrium phase which is misleading. Furthermore, talking about immunoediting without citing the seminal papers of Robert D Schreiber, who proposed the concept and provided experimental evidence of immunoediting is not acceptable.

  • We worked to clarify the introduction concerning immunoediting.  We added Innate and adaptive immunity to line 153 and cited Schreiber et al for he concepts  (ref 13, 14) (highlighted in green)

4. Figure 1 also is not clear and precise. Classification of “cancer cells” vs“immunogenic transformed cells” is vague. What exactly is the difference between them? Effector molecules and receptors listed in the boxes should be classified according to their type or functions. Abbreviations should be spelled out in the legend.

  • The critique is relevant and appreciated since we now realize the figure was not sufficiently clear to communicate the relevant information.  The mentioned figure has been extensively revised and reformatted to address the reviewers concerns. Effector molecules and receptors were classified into general groups based on broad characteristics. 

5. Tables 1, 2, 3, 4, and 5: cellular source of EVs should be included.

  • The cellular sources have been added to the tables.

6. Table 5: indicating “Plasma of patients” in the column “Cancer type” is not appropriate and should be replaced by the cancer types of patients from whom the plasma samples were collected.

  • The table 5 has been revised to address the critique

Reviewer 2 Report

The review by Najaflu et. al., titled “Tumor-derived exosomes in cancer immunoediting and their potential as onco- immunotherapeutics” discusses the role and implications of exosomes in cancer progression and their potential function as targets for immunotherapy.

While this is a topic of great interest for readers since there is an unmet need of novel therapeutic avenues for treatment of patients with cancer, especially those who develop resistance to existing drugs, in the last couple of years, a number of review articles have been published on this topic. However, this is a well written review and covers important aspects of the given topic satisfactorily.

My comments for the authors are listed below:

1.     Section 2.2 – authors have listed the mechanisms of action of TEX in modulating the activity and type of fibroblasts. My question is, do these mechanisms apply universally across different cancer types? The given references list certain cancer type only. I would suggest the authors either mention the cancer type wherein the mechanistic studies have comes from (as stated in segment 3.2.2) or add more studies in the references showing studies done in different cancer types.

2.     Figure 1 – Rearrange figure panel from a  b  c, as mentioned in the figure legend. The reverse direction of figure is confusing and distracting for readers.

3.     There are studies in which exosomes were used as drug carriers, e.g. Paclitaxel (PTX) and others. There should be a paragraph on such studies in section 4 and their clinical outcomes, if any.

4.     Are there currently any active clinical trials to check the role of TEXs for cancer immunotherapy? If so, a tabulated list with their current phase and status will be a useful addition.

5.     Majority of the references listed are reviews. It is strongly advised to cite the reference of original research article instead.

Minor comments –

1.     Sentence no. 96, 254 and others: Instead of using “like”, use “such as or example” since the factors are well defined in literature.

2.     Sentence no. 52 and 315-317 are unclear. It is best to write short and simple sentences for clarity. Sentences that take 3-4 lines must be avoided.

3.     Maintain uniform font throughout the article– correct line no. 8 and 305.

4.     Abstract line no. 33 – It is not a study but review of literature. Please make necessary changes.

Author Response

Reviewer 2

  1. Section 2.2 – authors have listed the mechanisms of action of TEX in modulating the activity and type of fibroblasts. My question is, do these mechanisms apply universally across different cancer types? The given references list certain cancer type only. I would suggest the authors either mention the cancer type wherein the mechanistic studies have comes from (as stated in segment 3.2.2) or add more studies in the references showing studies done in different cancer types.

  • We appreciate the useful critique. More references and new studies from different cancer types were added as also the cancer types mentioned in the text.

  1. Figure 1 – Rearrange figure panel from a àb à c, as mentioned in the figure legend. The reverse direction of figure is confusing and distracting for readers.

  • The Figure is completely reformed and revised to address both reviewers concerns.

  1. There are studies in which exosomes were used as drug carriers, e.g. Paclitaxel (PTX) and others. There should be a paragraph on such studies in section 4 and their clinical outcomes, if any.
  •  The issue was added as the last paragraph prior to the conclusion section of the article.

  1. Are there currently any active clinical trials to check the role of TEXs for cancer immunotherapy? If so, a tabulated list with their current phase and status will be a useful addition.

  • Not to our knowledge. Most of the current clinical trial studies using EVs for cancer are based on dendritic or other immune cells derived exosomes. TEV studies are still preclinical and since the unmodified TEVs from escape phase cancers are protumorigenic a variety of issues like impact on supporting metastasis needs to be considered.

  1. Majority of the references listed are reviews. It is strongly advised to cite the reference of original research article instead.

  • Many reviews were replaced with original articles the new original data articles are marked in yellow in the text citations for reviewers convenience. When in text citations are grouped, (example [5-7]) the whole group is yellow but sometimes not all the references are new data papers.

Minor comments –

  1. Sentence no. 96, 254 and others: Instead of using “like”, use “such as or example” since the factors are well defined in literature.

  • Those sentences are now revised

  1. Sentence no. 52 and 315-317 are unclear. It is best to write short and simple sentences for clarity. Sentences that take 3-4 lines must be avoided.

  • We agree with the reviewers stylistic critique. Those sentences are now revised and broken into 2 sentences.

  1. Maintain uniform font throughout the article– correct line no. 8 and 305.

  • The font has been corrected and is now uniform

  1. Abstract line no. 33 – It is not a study but review of literature. Please make necessary changes.

  • This has been revised

Reviewer 3 Report

The review by Najaflu et al. focused on the role of tumor derived EVs in the control of the immune system during tumor progression. The review is well written and, at difference from other review on the same topic, the Authors have considered  EV mechanisms and cargo during the 3 phases of tumor progression.

Minor concerns

The Authors should homogenize the term extracellular vesicles in the introduction (some time denoted as EVs other exosomes).

The following study must be included and discussed (PMID: 35398240).

Author Response

  1. The Authors should homogenize the term extracellular vesicles in the introduction (some time denoted as EVs other exosomes).

The terminology was also a concern of reviewers 1 and 2. We followed the nomenclature of the International Society for Extracellular Vesicles (as described in the introduction) the extracellular vesicles term replaced in the text.

  1. The following study must be included and discussed (PMID: 35398240).

The suggested article was added to the introduction (ref 18) and highlighted in green.

Round 2

Reviewer 1 Report

The manuscript has been thoroughly revised and my concerns are adequately addressed.